# A programmable qudit-based quantum processor

Yulin Chi[1,12], Jieshan Huang [1,12], Zhanchuan Zhang[1,12], Jun Mao [1], Zinan Zhou[1], Xiaojiong Chen[1], Chonghao Zhai[1], Jueming Bao[1], Tianxiang Dai [1], Huihong Yuan[1,2], Ming Zhang[3], Daoxin Dai[3], Bo Tang[4], Yan Yang [4], Zhihua Li[4], Yunhong Ding [5,6], Leif K. Oxenløwe [5,6], Mark G. Thompson [7], Jeremy L. O'Brien[8], Yan Li [1,9,10,11], Qihuang Gong [1,2,9,10,11] & Jianwei Wang [1,2,9,10,11]✉

Controlling and programming quantum devices to process quantum information by the unit of quantum dit, i.e., qudit, provides the possibilities for noise-resilient quantum communications, delicate quantum molecular simulations, and efficient quantum computations, showing great potential to enhance the capabilities of qubit-based quantum technologies. Here, we report a programmable qudit-based quantum processor in silicon-photonic integrated circuits and demonstrate its enhancement of quantum computational parallelism. The processor monolithically integrates all the key functionalities and capabilities of initialisation, manipulation, and measurement of the two quantum quart (ququart) states and multi-value quantum-controlled logic gates with high-level fidelities. By reprogramming the configuration of the processor, we implemented the most basic quantum Fourier transform algorithms, all in quaternary, to benchmark the enhancement of quantum parallelism using qudits, which include generalised Deutsch-Jozsa and Bernstein-Vazirani algorithms, quaternary phase estimation and fast factorization algorithms. The monolithic integration and high programmability have allowed the implementations of more than one million high-fidelity preparations, operations and projections of qudit states in the processor. Our work shows an integrated photonic quantum technology for qudit-based quantum computing with enhanced capacity, accuracy, and efficiency, which could lead to the acceleration of building a large-scale quantum computer.

[1] State Key Laboratory for Mesoscopic Physics, School of Physics, Peking University, 100871 Beijing, China. [2] Beijing Academy of Quantum Information Sciences, 100193 Beijing, China. [3] State Key Laboratory for Modern Optical Instrumentation, College of Optical Science and Engineering, Ningbo Research Institute, International Research Center for Advanced Photonics, Zhejiang University, 310058 Hangzhou, China. [4] Institute of Microelectronics, Chinese Academy of Sciences, 100029 Beijing, China. [5] Department of Photonics Engineering, Technical University of Denmark, 2800 Kgs. Lyngby, Denmark. [6] Center for Silicon Photonics for Optical Communication (SPOC), Technical University of Denmark, 2800 Kgs. Lyngby, Denmark. [7] Quantum Engineering Technology Labs, H. H. Wills Physics Laboratory and Department of Electrical and Electronic Engineering, University of Bristol, BS8 1FD Bristol, United Kingdom. [8] Department of Physics, The University of Western Australia, Perth 6009, Australia. [9] Frontiers Science Center for Nano-optoelectronics & Collaborative Innovation Center of Quantum Matter, Peking University, 100871 Beijing, China. [10] Collaborative Innovation Center of Extreme Optics, Shanxi University, 030006 Taiyuan, Shanxi, China. [11] Peking University Yangtze Delta Institute of Optoelectronics, Nantong 226010 Jiangsu, China. [12] These authors contributed equally: Yulin Chi, Jieshan Huang and Zhanchuan Zhang. ✉email: jww@pku.edu.cn

Natural quantum matters store rich multidimensional quantum information in a superposition of more than two electronic or mechanical modes. Engineering artificial multilevel quantum devices to mimic nature may allow fundamental innovations and technological advances. Recently, though the state-of-the-art qubit-based quantum technologies have demonstrated revolutionary milestones, e.g., loophole-free Bell tests[1,2], satellite-relayed quantum communications[3,4] and quantum computational advantages[5,6], qudit-based quantum technologies might be able to further enhance quantum capabilities as they are intrinsically consistent with our natural quantum systems. For example, entangled qudit states can strengthen the Bell nonlocality[7] and moderate the detection loophole[8]; distributing qudit states allows high-capacity noise-resilient quantum cryptography[9–11]; by mapping Hamiltonians into multilevel quantum devices, it can provide a direct solution for quantum simulations of complex molecular and physical systems[12–16]; more importantly, universal quantum computation with qudits is possible in both of the circuit models[17] and measurement-based models[18,19], requires less resource overhead in quantum error correction[20,21], and can improve the execution of quantum algorithms[22,23]. Heuristically, the exponential speed-up of many quantum algorithms is enabled by the quantum parallel evaluation of a function $f(x)$ for all input $x$ values simultaneously, as $\sum_x |x\rangle |f(x)\rangle$, where the $x$ input string is represented by a superposition of quantum states. The adoption of qudit as the basic quantum information unit in processing quantum algorithms[24] offers enhanced computational capacity that is represented by the size of the Hilbert space of $d^n$, where $n$ is the number of qudits and $d$ is the local size of each qudit. Moreover, it can lead to higher computational accuracy for example in implementing quantum Fourier transform algorithms such as Shor's fast factorisation[25] and phase estimation[26], in which the computational accuracy is determined by the size of auxiliary qudits. Processing the Kitaev's version of quantum Fourier algorithms[26–29] with qudits may allow further speed-up of quantum computing. These unique capabilities have strongly prompted the development of qudit-based quantum computing in universal models[17–19,30,31], and very recently in experimental controls of qudit states and logic gates in photonics[32], solid-state[15], trapped ion[33], and superconducting[34] platforms. In particular, photons are intrinsically multidimensional[35], enabling flexible and reliable encoding of qudits with their different degrees-of-freedom, e.g, path[36,37], frequency[38,39], spatial mode[40,41] and temporal mode[11,42]. Advances in the control of quantum photonic devices have recently allowed remarkable experimental progress. For example, multidimensional Greenberger-Horne-Zeilinger (GHZ) states and cluster states prepared in the frequency-bins and time-bins of two photons generated in a single microring resonator[43,44], have firstly shown enhancement in quantum computation by providing increased quantum resources and higher noise robustness compared to the qubit counterparts; An integrated photonic chip for the generation, manipulation and measurement of two-photon multidimensional Bell states has been demonstrated[36], while the scaling capability has been verified by the generation of multiphoton multidimensional GHZ states[45,46], and the realisation of single-qudit quantum teleportation[47,48]. Despite of these remarkable development of multidimensional quantum photonic technologies that mainly focus on the preparation and control of qudit states and gates, a monolithically integrated quantum device that is able to initialise, manipulate and analyze qudit states and gates is lacking. Furthermore, the programmability of quantum hardware presents the major enabling capability of quantum computing technologies. For example, several milestones in qubit-based quantum computing have been all realised in programmable quantum devices of photons[49,50], trapped ions[51,52], superconductors[5,6] and semiconductors[53]. However, limited to the best to our knowledge, such a qudit-based quantum computing device that can be fully reconfigured and reprogrammed to implement different tasks has not been realised to date, in any quantum system. Likely, it requires an integrated platform[35,54,55], capable of initialising, manipulating and measuring qudit states and gates, in a fully controllable and highly programmable manner. Realising a programmable qudit-based quantum processor therefore presents a significant step to transition the technological advances of controlling qudit states and logic gates to the implementations of quantum tasks and quantum computational algorithms, in $d$-ary.

In this work, we demonstrate a programmable qudit-based quantum processing unit ($d$-QPU) on a large-scale silicon-photonic quantum chip. The initialisation, manipulation and measurement of arbitrary single-qudit and two-qudit states, and multi-value quantum-controlled logic gates can be implemented on the single $d$-QPU chip. Such a fully monolithic integration of all necessary functionalities allows the implementation of a top-down hierarchy of programmable qudit-based quantum computation, as shown in Fig. 1. Different quantum tasks and computational algorithms are implemented, all in quaternary, by recompiling the qudit logic circuit in the software level, and then executing the circuit by reprograming the configurations of the $d$-QPU chip in the hardware level. We then benchmark the enhancement of quantum computational parallelism, by per-

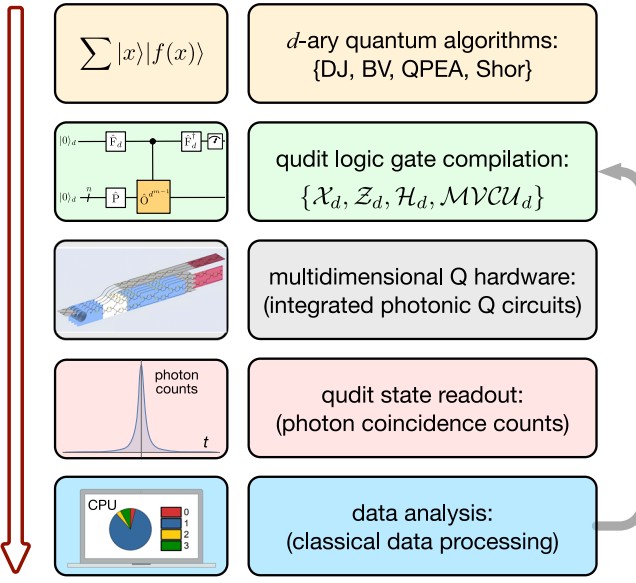

**Fig. 1 The top-down hierarchy of qudit-based quantum computation.** Users can define different quantum tasks and implement different quantum algorithms in $d$-ary, e.g., the generalised Deutsch-Jozsa (DJ), Bernstein-Vazirani (BV), quantum phase estimation (QPEA) and Shor's fast factorization algorithms. In the software level, a multi-qudit quantum logical circuit for executing the algorithm is compiled with single-qudit gates (e.g, $Z_d$, $X_d$ and $H_d$) and multi-value controlled-unitary (MVCU$_d$) gates. In the hardware level, the logical circuit is physically implemented by an integrated photonic quantum device, i.e, the programmable $d$-QPU, and the gate operations are realised by optical waveguide devices, such as entangled photon sources, phase shifters, beamsplitters and interferometers. Multiple quantum tasks and algorithms can be executed, without the need of altering the device, only by reprogramming the configurations of waveguide circuits. The outcome of the hardware is given by photon coincidence counts, which are recorded and analysed by classical electronics and classical computer. Experimental outcomes can be feed-forwarded into the $d$-QPU for the implementation of Kitaev's version of quantum Fourier transform algorithms.

forming the generalised Deutsch-Jozsa and Bernstein-Vazirani algorithms, quaternary phase estimation and order finding algorithms. Our results show a proof-of-principle demonstration of qudit-based quantum computer with integrated optics, that allows improvement of the capacity, accuracy and efficiency of quantum computing.

## Results

**Scheme of multiqudit quantum processor.** Figure 2 shows the core of a multiqudit processor, i.e, the multiqudit multi-value controlled logic gate, which is realised by the following three steps: generation of the multiphoton multidimensional Greenberger-Horne-Zeilinger entangled state $|GHZ\rangle_{n+1,d}$[45,46], which enables the entangling operations between the multiqudit states; Hilbert space expansion of each qudit in $y$-register to form an entire space of $d^{2n}$, that locally allows individual and arbitrary single-qudit operations[56]; coherent compression of the entire state back to a $d^n$ space[57]. These sequences of operations result in a multiqudit multi-value controlled-unitary (MVCU) gate as $\frac{1}{\sqrt{d}}\sum_{j=0}^{d-1}|k_j\rangle \otimes \prod_{i=1}^{n} O_{i,j}|\phi\rangle_i$, where $|k_j\rangle$ in the auxiliary $x$-register presents the logical state in the $j$-th mode (for simplicity it is denoted as $|j\rangle$), and $O_{i,j}$ in the data $y$-register refers to an arbitrary local operation on the qudit state $|\phi\rangle_i$ that is initialised by the $p_i$ qudit generator. Such multiqudit MVCU gate works with a $(1/d)$ success probability regardless of $n$ (see Supplementary Note 3 and Supplementary Fig. 1). The quantum circuits in Fig. 2a, b provide a scheme of implementing multiqudit quantum Fourier algorithms in the scalable Kitaev's framework[26–29].

Figure 2c illustrates the integrated photonic quantum circuits for a two-ququart version of qudit-based quantum processing unit ($d$-QPU). It was fabricated in silicon using the complementary metal-oxide-semiconductor (CMOS) process with the 248nm deep ultraviolet lithography (see a device image in Fig. 2d). The processor allows the generation of a path-encoded two-ququart entangled state of $|GHZ\rangle_{2,4}$ (i.e., the 4-dimensional generalised Bell state of $|Bell\rangle_4$), by a coherent excitation of four integrated spontaneous four-wave-mixing (SFWM) sources. It is followed by the sequences of processes of "space expansion–local operation–coherent compression" for the realisation of $d$-QPU, see Fig. 2b. The $d$-QPU chip monolithically integrates the core capabilities and functionalities, including arbitrary single-ququart preparation ($P$), arbitrary two-ququart MVCU operation (that presents a $d$-ary generalisation of two-qubit controlled-unitary operation), and arbitrary single-ququart measurement ($M$). Though on-chip generation, manipulation and measurement of entangled qudit states have been reported[36], this work demonstrate the key abilities to initialize, manipulate, and analyze qudit states and gates in a fully reconfigurable and reprogrammable manner, providing a major technological advance for qudit quantum computing. In Fig. 2d it shows one of the largest-scale programmable quantum photonic chip having 451 photonic components, including 116 reconfigurable phase-shifters (see their characterisations in Fig. 2c insets). The two-photons detection rate at the magnitude of $10^3/s$ was measured in the two-ququart device, which is six orders higher than that in a four-qubit device (note the detection rate depends on the performance and loss of the quantum devices as well as their pumping and measurement apparatuses)[58]. Details of device fabrication, state evolution and experimental setup are provided in Supplementary Notes 1 and 3.

**Characterisation of $d$-ary multi-value controlled-unitary gates.** Before reporting experimental results, we first define classical statistic fidelity ($F_c$) and quantum state (process) fidelity ($F_q$), used in this work to quantify the qudit states, logic gates and algorithm implementations. The $F_c$ is defined as $(\sum_i \sqrt{p_i q_i})^2$, where $p_i$, $q_i$ are theoretical and measured distributions, respectively; the state $F_q$ is defined as $(\text{Tr}[\sqrt{\sqrt{\rho_0} \cdot \rho \cdot \sqrt{\rho_0}}])^2$, where $\rho_0$, $\rho$ are ideal and measured states, respectively; the process $F_q$ is defined as $\text{Tr}[\chi_0 \chi]$, where $\chi_0$, $\chi$ are ideal and reconstructed process matrices, respectively.

We first characterised the single-ququart and two-ququart logic gates. As examples, two single-ququart gates are characterised: the generalised $d$-level Pauli-$X_d$ gate that is defined as $X_d|k_i\rangle = |k_{(i\oplus_d 1)}\rangle$ where $\oplus_d$ is addition module of $d$, and the $d$-level quantum Fourier gate $F_d$ that transforms the computational basis of $|k_i\rangle$ to the Fourier basis $|f_i\rangle$ of $\frac{1}{\sqrt{d}}\sum_{j=0}^{d-1}\omega^{ij}|j\rangle$ where $i,j \in N_d$ and $\omega = \exp(\mathrm{i}\frac{2\pi}{d})$. When $d$ is two, they return to the standard Pauli and Fourier (Hadamard) gates for qubits. In Fig. 2c inset, it shows the measured mean $F_c$ of 0.988(13) for the five $X_4$ gates and 0.967(19) for the five $F_4$ gates, where the values in parentheses are uncertainty from photon statistics. Next, we characterised the two-qudit entangling gate:

$$\text{MVCU}\,|x\rangle|y\rangle = |x\rangle O^x|y\rangle, \tag{1}$$

where $O$ can be arbitrarily operated[59] on the $|x\rangle$ and $|y\rangle$ registers. Notably, the MVCU gate presents a coherent entanglement between the auxiliary $x$-register and the data $y$-register. The processing of $d$-ary quantum algorithms relies on the multiple path interference in the $d$-dimensional Fourier gate to obtain the desired solutions. Such coherent superposition of qudits ensures quantum parallelism, that is function evaluations for multiple inputs are executed in parallel. The MVCU is thus a core logic enabling the quantum parallel evaluation of the function. For example, as the $d$-ary generalisation of the CNOT gate[24], the MVCX$_d$ gate allows the creation of a complete set of four-level Bell states $|\Psi\rangle_{i,j}$ defined as $\frac{1}{2}\sum_{m=0}^{3}\omega^{mi}|m\rangle|m\oplus_d j\rangle$, by inputing the $|f_i\rangle \otimes |k_j\rangle$ states into the logic, $i,j = 0,1,2,3$. Figure 3a shows the reconstructed $|\Psi\rangle_{12}$ state, and Fig. 3c shows measured $F_q$ for the 16 Bell states with an averaged fidelity of 0.967(31). The state matrices ($\rho$) represented as a linear combination of Gell-Mann matrices were reconstructed by implementing compressed sensing quantum state tomography techniques[60]. In addition, a fully product state was created in Fig. 3b, given an input of $|f_0 f_0\rangle$. Figure 3d shows the experimental process matrix ($\chi$) of the MVCX$_d$ gate, by performing quantum process tomography with a full set of 256 state tomographic measurements[61], and a process fidelity $F_q$ of 0.952 was obtained. We then characterised the MVCZ$_d$ gate ($Z_d$ is the generalised $d$-level Pauli-$Z_d$ gate) transforming $|x\rangle|y\rangle$ to $|x\rangle\omega^{xy}|y\rangle$, and the MVCH$_d$ gate where $H_d$ is the $d$-level Hadamard gate with elements of $\frac{1}{\sqrt{d}}(-1)^{i\odot j}$ ($i\odot j$ is the bitwise dot product, see Supplementary Note 2). Instead of performing full process tomography, we adopted an efficient characterisation by using complementary classical fidelity[62]. Figure 3e–j show measured input-output truth tables and their classical fidelity ($F_{c1}$, $F_{c2}$) for the MVCU in two complementary {base I, base II}, from which the complementary classical fidelity is upper and lower bounded by $[F_{c1} + F_{c2} - 1, \text{Min}(F_{c1}, F_{c2})]$.

**Experimental implementation of $d$-ary Deutsch's algorithms.** The class of Deutsch's algorithms well identify quantum parallelism. A generalised $d$-ary Deutsch-Jozsa algorithm can determine whether a multi-value function $f: \{0,1,...,d-1\}^n \rightarrow \{0,1,...,d-1\}$ is constant or balanced by a single query of a quantum oracle[63]. Classically, it however requires $d^{n-1} + 1$ queries. The quantum circuit performing $f(x)\oplus_d y$ is shown in Fig. 4a. In the case of $d = 2$, it returns to the original binary Deutsch-Jozsa[64]. We implemented the ququart Deutsch-Jozsa algorithm on the $d$-QPU for the case of $n = 1$ and $d = 4$. Figure 4b–h show the measured probability distributions of the $x$-register in the computational basis, when the multi-value

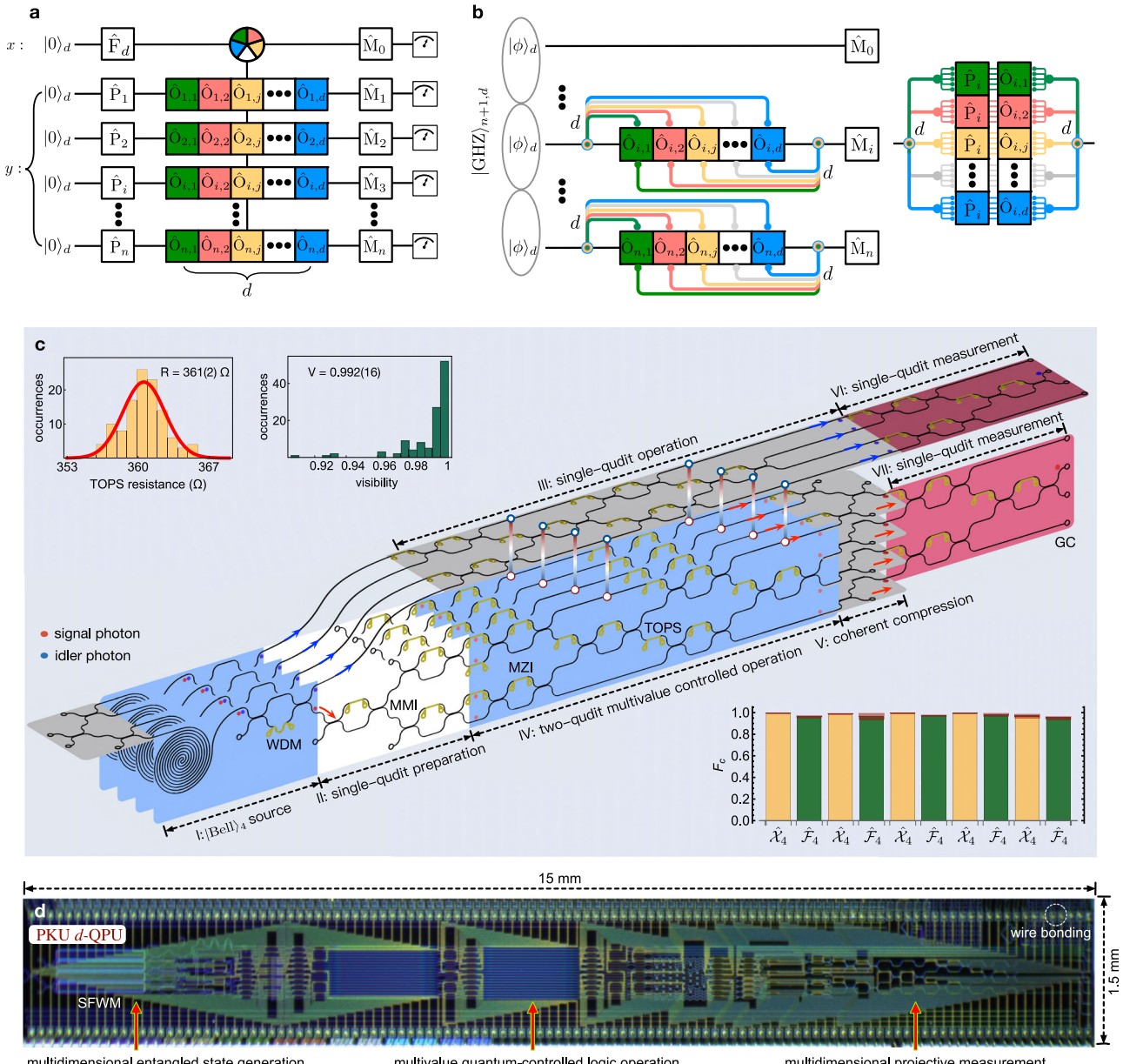

**Fig. 2 A qudit-based programmable quantum processing unit in a photonic integrated circuit chip. a** Quantum circuit, and **b** physical implementation of the multiquit QPU. It bases on multiphoton multidimensional entanglement of $|GHZ\rangle_{n+1,d}$, where $n+1$ is the number of photonic qudits and $d$ is the local dimensionality of each qudit. $P_i$ is an arbitrary single-qudit gate; $F_d$ is a generalised $d$-level Fourier gate; $M_i$ is an arbitrary single-qudit projector; $O_{i,j}$ ($i=1,...,n, j=1,...,d$) is an arbitrary single-qudit logic gate that is locally performed on the $i$-th qudit of the $y$-register, and the $O_{i,j}$ gates are coherently entangled with the $x$-register state. The process of "space expansion–local operation–coherent compression" results in the multiquit entangling gate, with a success probability of $1/d$, independent on $n$. **c** The simplified schematic of a two-ququart $d$-QPU: (I) generation of four-level entangled state in an array of four integrated identical SFWM sources; (II) Hilbert space expansion and arbitrary single-qudit preparation of the $y$-register state; (III) arbitrary single-qudit operation of the $x$-register state; (IV) arbitrary single-qudit operation (loading in the four layers) of the $y$-register state, in which the operations are coherently entangled with the $x$-register state, thus forming the MVCU entangling gate, where the state-gate entanglement is indicated by the four colourful links; (V) coherent compression of Hilbert space by an indistinguishable erasure of spatial information; (VI) and (VII) arbitrary single-qudit projective measurement in the $x$ and $y$ registers. Insets: left top, measured resistance of all thermal-optic phase shifters (TOPSs); measured interference visibility of all 2-dimensional Mach-Zehnder Interferometers (MZIs); bottom right, measured classical statistic fidelities ($F_c$) for the Pauli $X_4$ gate with a mean of 0.988(13) and Fourier $F_4$ gate with a mean of 0.967(19). **d** A microscopy image of the $d$-QPU chip. It monolithically integrates 451 optical components, including 4 SFWM sources, 116 reconfigurable TOPS, 131 multimode interferometer (MMI) beamsplitters, 4 wavelength-division multiplexing (WDM) filters, 156 waveguide crossings and 40 grating couplers (GC). The $d$-QPU chip is wire bounded and can be flexibly controlled by classical electronics, and can be reliably reprogrammed and reconfigured to benchmark a spectrum of different quaternary quantum algorithms.

function is chosen as constant (see Fig. 4b) and balanced (see Fig. 4c–h), respectively. The $d$-QPU thus determines whether $f$ is constant or balanced, and the fidelity $F_c$ of 0.967(2) was measured to quantify its success probability. Notably, the measured distributions in Fig. 4b, c, h, i are fully distinguishable. These imply an interesting capability of computing a close expression for an affine function $f$: $A_0 \oplus A_1 x_1 \oplus ... \oplus A_n x_n$. That presents the $d$-ary generalisation of the Bernstein-Vazirani algorithm[65], whose task is to compute the $d$-ary

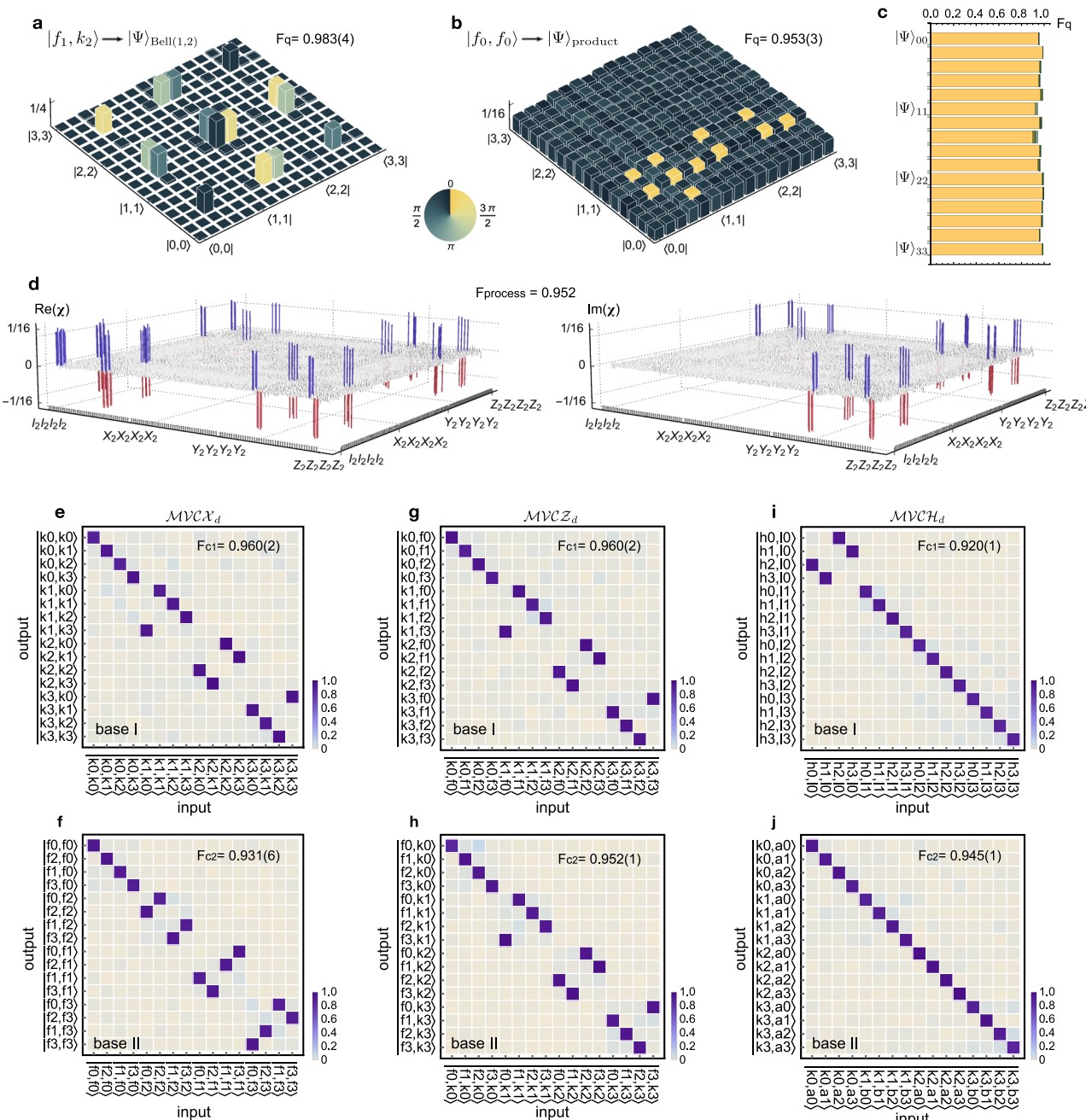

**Fig. 3 Characterisation of quaternary multi-value controlled-unitary logic operations. a**, **b** Measured density matrices ($\rho$) for a four-level maximally entangled Bell state and a fully product state. Column heights (colours) represent absolute values $|\rho|$ (phases $\text{Arg}(\rho)$) of the elements. Quantum state fidelity $F_q$ was measured to be 0.983(4) and 0.953(3), respectively. **c**, Measured quantum state fidelities for a complete set of four-level Bell states $|\Psi\rangle_{i,j}$, $i, j = 0,1,2,3$. The generalised Bell states are created by operating input states of $|f_i\rangle \otimes |k_j\rangle$ in the MVCX$_d$ gate. Shaded areas atop bars refer to $\pm 1\sigma$ error bars. The $F_q$ values in **a–c** were estimated by Monte Carlo the photon counts with photon Poissonian statistics. **d** Reconstructed process matrix ($\chi$) of the MVCX$_d$ gate. It was measured by quantum process tomography with in total 256 quantum state tomographic measurements. We obtained the quantum process fidelity of 0.952, that is defined as $\text{Tr}[\chi_0\chi]$, where $\chi_0$ is the ideal matrix. The $\chi$ matrix is represented in the standard identity and Pauli basis $\{I_2, X_2, Y_2, Z_2\}$. Blue and red colours are used to improve the clarity. **e–j** Measured truth tables (normalised photon counts) for three MVCU logic gates in two complementary bases {I, II}: **e**, **f**, a multi-value controlled-$X_d$ (MVCX$_d$) gate; **g**, **h**, a multi-value controlled-$Z_d$ (MVCZ$_d$) gate; **i**, **j**, a multi-value controlled-$H_d$ (MVCH$_d$) gate. The definitions of basis are given as: computational basis $|k_i\rangle$; Fourier basis $|f_i\rangle$; Hadamard basis $|h_i\rangle$; basis $|l_i\rangle$ is another eigenstate of the Hadamard and $|a_i\rangle$ and $|b_i\rangle$ are given by rotations, which are provided in Supplementary ($i = \{0, 1, 2, 3\}$). Classical statistic fidelities ($F_{c1}$, $F_{c2}$) are measured, which are adopted to estimate the lower and upper bound of the complementary classical fidelity: [0.891(2), 0.931(1)] for the MVCX$_d$, [0.912(2), 0.952(1)] for the MVCZ$_d$, and [0.865(1), 0.920(1)] for the MVCH$_d$. In **e–j**, the probability distributions are colour coded (key is provided at the right bottom). The values in parentheses of $F_c$ and $F_q$ refer to $\pm 1\sigma$ uncertainty from photon statistics.

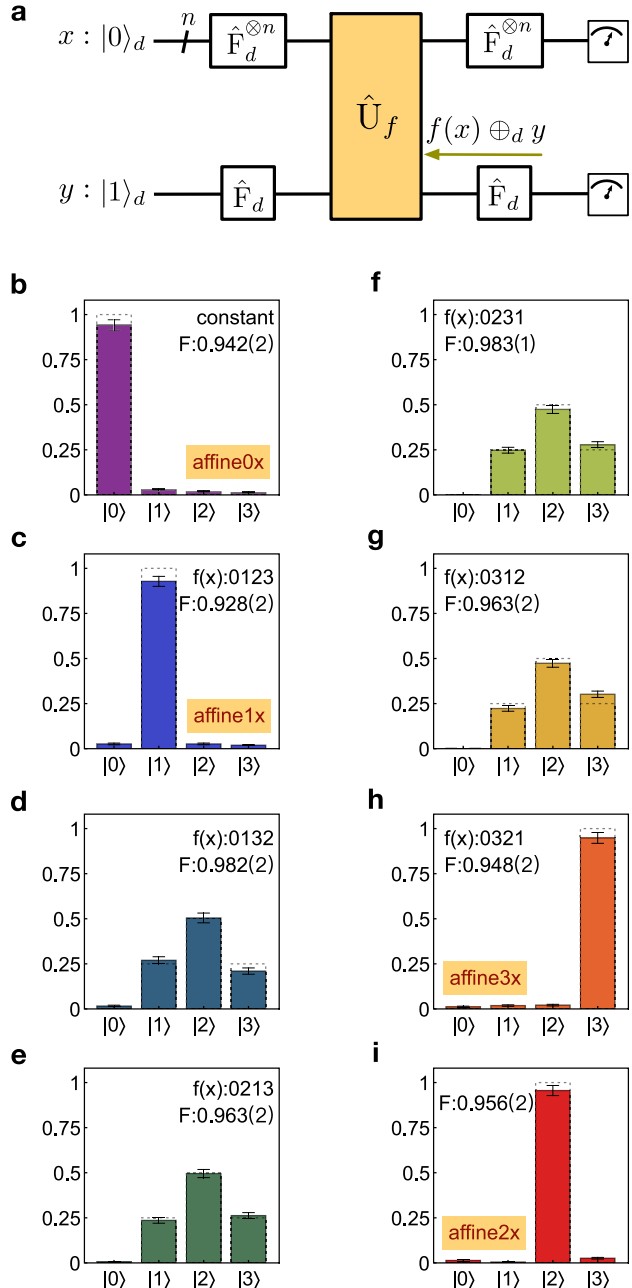

**Fig. 4 Implementations of generalised Deutsch-Jozsa and Bernstein-Vazirani algorithms in quaternary. a** Quantum logical circuit for implementing the *d*-ary Deutsch-Jozsa and Bernstein-Vazirani algorithms. This circuit can be implemented by the scheme in Fig. 1a, b with an exchange of the *x* and *y* registers. The task of the *d*-ary Deutsch-Jozsa algorithm is to determine an unknown multi-value function *f*: {0, 1,..., *d*−1}$^n$ → {0, 1,..., *d* − 1} is either constant or balanced, while that of the *d*-ary Bernstein-Vazirani algorithm is to compute the close expression of a multi-value affine function *f*: $A_0 ⊕ A_1 x_1 ... ⊕ A_n x_n$, using only a single call of quantum oracle. When *d* equals to 2, the two algorithms return to the original Deutsch's algorithms. The key part is the implementation of $f(x) ⊕_d y$ by the MVCU gate. The outcome of the algorithms is measured in the computation basis of the *x*-register states. **b–i** Measured probability distributions (normalised coincidence counts) of the *x*-register in the computational basis. Results in **b–h** demonstrate that the *d*-ary Deutsch-Jozsa algorithm allows the determination of whether *f*(*x*) is constant (**b**) or balanced (**c–h**). Results in **b**, **c**, **i**, **h** show the *d*-ary Bernstein-Vazirani algorithm can determine the expression of affine functions *f*: **b**, *f*(*x*) is constant and $A_1$=0; **c**, *f*(*x*) is affine and $A_1$=1; **i**, *f*(*x*) is affine and $A_1$=2; **h**, *f*(*x*) is affine and $A_1$=3; Dotted boxes in (**b--i**) refer to theoretical probability distributions. Experimental probability distributions (coloured bars) are obtained from photon coincidences, which are accumulated by 20s per measurement. The classical fidelity $F_c$ presents the success probability of each measurement. In order to make the small error bars visible in the plots, they are plot by ± 3$σ$. The values in parentheses refer to ± 1$σ$ uncertainty. All error bars are estimated from photon Poissonian statistics.

coefficients $A_i$. The output state of the *x* register can be derived as $ω^{-A_0} |A_1, A_2, ..., A_n⟩$, where the $|A_1, A_2, ..., A_n⟩$ state can be directly read out in its computational basis ($A_0$ is lost as a global phase).

From the experimental results in Fig. 4b, c, h, i one can therefore determine the multi-value function with $A_1$= {0, 1, 2, 3}, respectively, by a single query of the oracle. Details of the generalisation of the Deutsch's algorithms are provided in Supplementary Note 4.

**Benchmarking of *d*-ary phase estimation and order finding.** Quantum phase estimation and order finding are two of the most featured quantum Fourier transform ralgorithms, that are essential to molecular simulation[66] and fast factorisation[25]. Kitaev's scalable implementation of both algorithms (in binary)[26–29] has been reported in several leading quantum platforms[67–72]. The remarkable idea is to replace the 2*n* qubits by

a single qubit in the auxiliary *x*-register, but at the expense of repeating m-sequences of single-qubit measurement and single-qubit feedforwarded operation, see quantum circuits in Fig. 5a. In Kitaev's phase estimation and order-finding algorithms, the computational capacity is determined by the number of *n*-qubits in the *y*-register, and the computational accuracy is determined by the number of *m*-sequences in the *x*-register. In this respect, one can see processing quantum algorithms with qudits results in nontrivial advantages: a $\log_2(d)$ larger computational capacity, and $\log_2(d)$ higher computational accuracy or $\log_2(d)$-less computational steps to achieving the same precision, as shown in Supplementary Fig. 3b, which could be important to quantum computers with limited coherence time.

In the quantum phase estimation, we aim to compute the eigenphase $ϕ$ of an unitary as $O|ψ⟩ = e^{i2πϕ}|ψ⟩$, given the eigenstate of $|ψ⟩$. The eigenphase of $ϕ$ can be described in *d*-ary as 0.$ϕ_1 ϕ_2...ϕ_{m-1} ϕ_m$, where *m* denotes iterative steps determining the approximation accuracy[26,29], and each dit of the phase is in [0, 1,..., *d* − 1][67,69,73]. We take the *s*-th step as an example (see quantum circuit in Fig. 5a). We prepare an input state of $\frac{1}{\sqrt{d}} \sum_{j=0}^{d-1} |j⟩|ψ⟩$ and perform the MVCU gate, that results in a state of $\frac{1}{\sqrt{d}} \sum_{j=0}^{d-1} e^{ij2π(0.ϕ_s ϕ_{s+1}...ϕ_m)}|j⟩|ψ⟩$. Then, the *x*-register qudit state is feed-forwardly rotated around the Pauli $Z_d$ basis as diag[1, $e^{i2πθ_s}, ..., e^{i2π(d-1)θ_s}$], where the rotation angle $θ_s$ of − 0.0$ϕ_{s+1} ϕ_{s+2}...ϕ_m$ is given by previous measurement outcomes. Remarkably, implementing an inverse $F_d$ in the *x*-register returns an output state as $|ϕ_s⟩ = |s⟩$ (see Supplementary Note 5). Measuring the *x*-register in the computational basis of $|s⟩$ therefore allows the extraction of the *s*-th dit of the dit expansion. The algorithm iteratively computes all *m* dits of the eigenphase backwardly, in which, notably, each dit is once estimated with *d*-ary accuracy. Figure 5b–d report measured eigenphases of 4-dimensional unitary matrices by quaternary phase estimation. We estimated the four eigenphases for three logic gates, i.e., a phase gate $Z_4$, a Fourier gate $F_4$

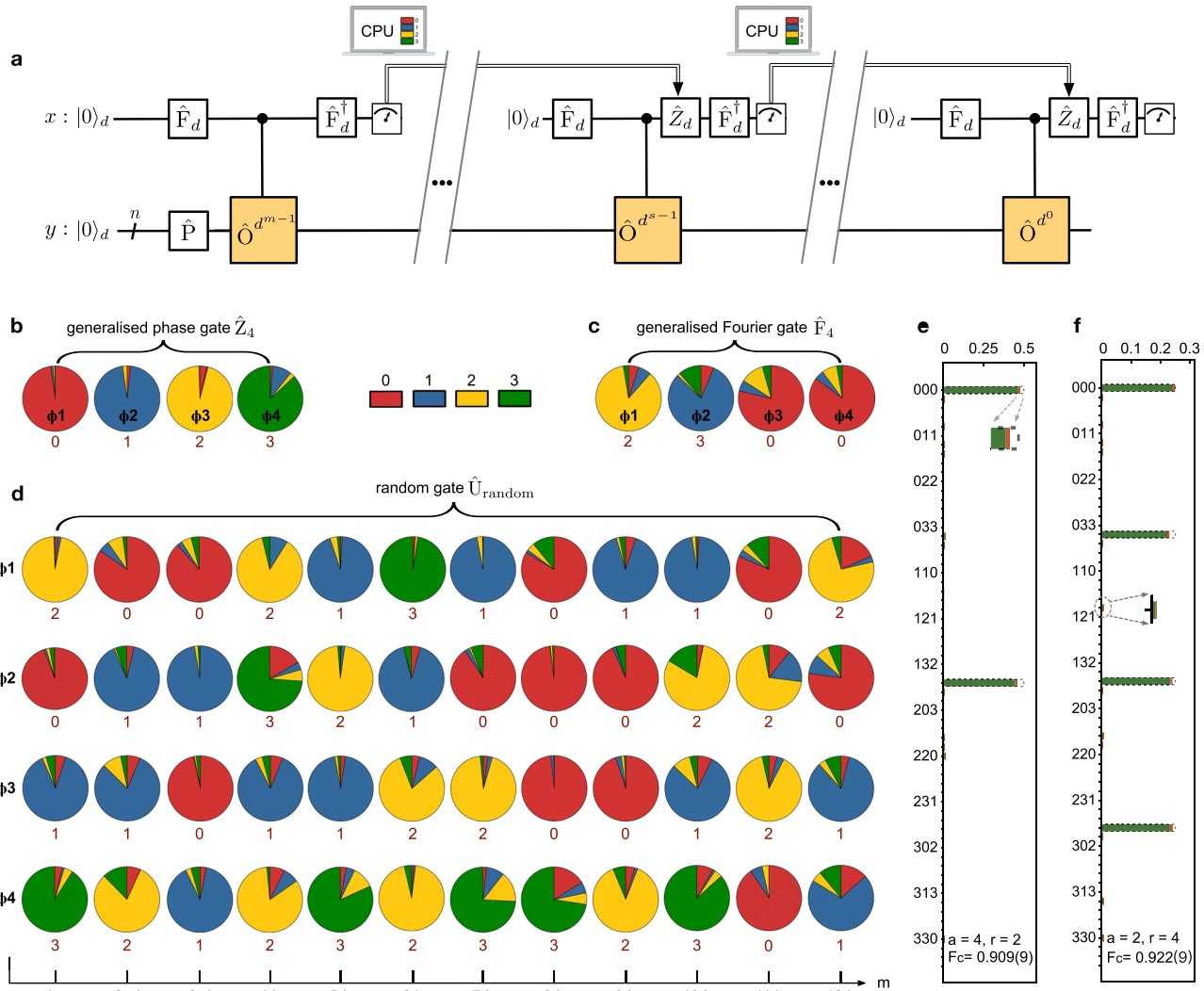

**Fig. 5 Implementations of quaternary quantum phase estimation and order-finding. a** Their quantum logical circuit for implementing Kitaev's scalable approaches. For the $d$-ary phase estimation, the task is to compute the eigenphase $\phi$ of a unitary $O$ given its eigenstate of $|\phi\rangle$. For the $d$-ary order-finding, the task is to find the order of a function as $(a^r \bmod N) = 1$. The $x$-register single-qudit state is initialised by the Fourier gate $F_d$; the $y$-register is prepared in the $|\phi\rangle$ eigenstate ($|0\rangle_d^{\otimes n}$ state) for phase estimation (for order-finding). The $F_d^\dagger$ terminates the $x$-register to output the desired solution in the computational basis. In the $s$-th step, the $Z_d$ rotation is added with a feedback angle of $\theta_s = -0.0\phi_{s+1}\phi_{s+2}...\phi_m$, that is determined by previous measurements. The algorithm is iterated $m$ times -- each step returns 1 dit result with $d$-ary accuracy, to obtain a $m$-dit estimation of the eigenphase of a unitary or the order of a function. **b**–**d** Measured probability pie-distributions of the four eigenphases ($\phi_1, \phi_2, \phi_3, \phi_4$) for three different unitary matrices, using the quaternary quantum phase estimation: **b** a generalised phase gate $Z_4$ as $\mathrm{diag}[1, e^{i2\pi\theta}, e^{i2\pi2\theta}, e^{i2\pi3\theta}]$ where $\theta = 1/4$; **c** a generalised Fourier gate $F_4$; **d** a random gate $U_{\text{random}}$ (see form in Supplementary Note 5). Coloured sectors represent the experimental outcomes of $\{0,1,2,3\}$ for each iteration, measured in the computational basis of $\{|0\rangle, |1\rangle, |2\rangle, |3\rangle\}$, respectively. The measured dominating sector is used to obtain every dit of the eigenphases; theoretical values for each dit are provided under the pies. The eigenphases are backwardly computed from the least significant dit from $m = 12$ to 1. **e**, **f** Measured probability distributions for the quaternary order-finding algorithm with a setting of $a = 4$ and $a = 2$, respectively. From the distributions, the order of $r = 2$ and $r = 4$ are experimentally computed with a 3-quart resolution (equivalent to 64-level), and with a classical statistic fidelity ($F_c$) of 0.909(9) and 0.922(9), respectively. The order-finding together with classical algorithm allows the factorisation of $15 = 3 \times 5$. Errors ($\pm 1\sigma$) arising from photon Poissonian noise are indicated as red shaded caps. Dashed lines refer to theoretical predictions. Experimental probability distributions in **b**–**f** are calculated from photon coincidences, which are accumulated by 20s per measurement.

and a randomised gate $U_{\text{random}}$ (see their explicit forms in Supplementary). Each pie chart presents one dit measurement outcomes, and the area of each coloured sector denotes measured probability distributions in the computational basis of $\{|0\rangle, |1\rangle, |2\rangle, |3\rangle\}$, respectively. In Fig. 5b, c, the eigenphases of $Z_4$ and $F_4$ gates can be described by a single dit. Figure 5d shows the computed eigenphases of the $U_{\text{random}}$ gate with an accuracy of 12 dits, by running the algorithm with a number of 12 interactions on the $d$-QPU. Instead, in the qubit-based device, achieving the same

computational accuracy of $\pm 4^{-12}$ requires a number of 24 computational interactions. And the achieved computational accuracies of 12 quarts are sufficient for the calculation of molecular eigenenergies[67,69]. In Fig. 5, it shows experimental data are in good agreement with theoretical predictions (indicated under each pie).

The task of quantum factoring is to efficiently compute the prime factors $p$ and $q$ from an integer $N$[25]. It can be reduced to the task of finding the order $r$ of $a$ module $N$, i.e., by computing a function $f$: $a^r \bmod N = 1$ ($a$ is a co-prime of $N$), and with a high

probability it returns a factor as $gcd(a^{r/2} \pm 1, N)$, where $gcd(\alpha, \beta)$ refers to the greatest common divisor of $\alpha$ and $\beta$. As the order-finding is just the phase estimation of a unitary having the eigenphases of $s/r$, $s \in [0, r-1]$, one can directly adopt the $d$-ary phase estimation to determine the order of $r$ in the $d$-ary format. It can be considered as a generalisation of $d$-ary order-finding by adopting Kiteav's iterative techniques[70–72] (see details in Supplementary Note 6). We then reprogrammed the $d$-QPU to implement the order-finding in quaternary. The $a \in [0, r-1]$ satisfying $gcd(a, 15) = 1$ is randomly chosen. We chose $a = 4$ and 2 as examples, and set the unitary of the MVCU gate as $\{I_d, X_d, I_d, X_d\}$ and $\{I_d, X_d, X_d^2, X_d^3\}$, respectively, where $I_d$ is the $d$-mode identity. In our experiment, the order-finding algorithm was iteratively implemented by three steps, and each step returns quaternary outcomes in the computational basis, resulting in the 3-quart (64-level) computational accuracy of the $s/r$ eigenphase. Figure 5e, f show the measured output probabilities of the $x$-register in the computational basis of $|ijk\rangle$, $i, j, k = 0, 1, 2, 3$. Classical statistic fidelities $F_c$ of 0.909(9) and 0.922(9) were obtained in comparison with theoretical distributions, showing successful estimations of the order of $r = 2$ (Fig. 5e) and $r = 4$ (Fig. 5f), respectively. The $d$-QPU thus finds the order with double-enhanced computational accuracy; alternatively speaking, it executes the task twice faster than a qubit-QPU, given the same estimation precision. The order-finding algorithm together with classical processing using the continued fraction algorithm returns the factor of $gcd(a^{r/2} \pm 1, N) = (3, 5)$. Implementing the $d$-ary algorithms in the $d$-QPU can therefore find the order of a function and compute the eigenphase of a unitary, with a $\log_2(d)$-faster computational speed.

## Discussion

We have reported a proof-of-principle experimental demonstration of a programmable qudit-based quantum processor in photonic integrated circuits, and implementations of several generalised $d$-ary quantum Fourier transform algorithms in the $d$-QPU chip. In agreement with the references[17–19,24,43–46], our experimental results show that qudit-based quantum computation with integrated photonics can enhance quantum parallelism in terms of the computational capacity, accuracy and efficiency, in comparison with its qubit-based quantum computing counterpart. The computational capacity of the two ququart quantum processor is equivalent to that of a four-qubit processor, thus allowing the implementations of the Deutsch's algorithms for a function with longer-string. Keeping the same number of photons $n$ but encoding each qudit in a dimension $d$, not only gives a larger Hilbert space[74], but also significantly improve the detection rate of photons[43,44]. We obtained the detection rate of about 6 orders brighter than that in another device with the same Hilbert space[58]. More analysis is provided in Supplementary Fig. 3. Moreover, multiple parallel evaluations of the function and multiple path interference in the $d$-ary quantum Fourier gate, allow the enhancement of the computational efficiency and speed up of the determination of desired solutions. In the implementations of Kiteav's phase estimation and factorisation, a number of $\log_2(d)$-less iterations are needed in the qudit processor, i.e., a $\log_2(d)$ times speed up of quantum computation, compared with the qubit ones, given the same computational accuracy (see Supplementary Fig. 3b).

As the multi-value quantum controlled gates are the result of the entanglement in the generation stage and the gates are instead local operations that steer the state to collapse in the desired outputs, our scheme can be straightforwardly generalised to multi-qudit quantum computation. Its scalability is naively dependent on the number ($n$) of qudits and the dimensionality ($d$) of each qudit. Regarding the dimension of units, though the ququart states are implemented as an example in this work, it is straightforward to extend to a larger-$d$ device[36], which can be fabricated using the same CMOS fabrication techniques. Remarkably, this entanglement-assisted $d$-QPU scheme works with a success probability of $1/d$ regardless of $n$ (Supplementary Note 3). The scaling of $d$-QPU therefore strongly relies on the generation of the qudit GHZ entangled states. Combing the state-of-the-art technologies, including the techniques of generating multi-photon qudit GHZ states[45,46], on-chip high-fidelity control of qudit states[36], high-quality photon-pair sources[75,76], low-loss fibre-chip interface[75,77], and large-scale quantum integration[57], we estimate a 10-photons $d$-QPU is achievable in near term. Its further scaling requires high-efficiency heralded multiplexing photon-pairs sources[78] and multiplexing qudit GHZ generators[31]. That being said, given the efficient generation of the multi-photon multi-qudit GHZ states, the $d$-QPU scheme is scalable. Calculations and analysis are provided in Supplementary Note 9 and Supplementary Fig. 3d. Moreover, when scaling up the $d$-QPU, an interesting concern is the required resources, in particular the number of classical controls. As an example, let us consider a processor with one qudit in the auxiliary register and $n$ qudits in the data register (see Fig. 2a). It requires a number of $(n+1)$ single-qudit generators for state preparation, $(nd)$ local single-qudit operators for multi-qudit MVCU operation, and $(dn+1)$ single-qudit projectors. The physical resources, i.e, the number of phase-shifters, scale with $(d^2 - d)$ for the qudit operators[50,59], and $2(d-1)$ for the qudit generators and projectors[36], as shown in Supplementary Note 9 and Supplementary Fig. 2. Importantly, the required resources for classical controls scale polynomially with the number of particles. In Supplementary Fig. 3c, it is shown that thousands of phase-shifters are required for a 10-photon $d$-QPU. This large amount of phase-shifters can be individually addressed and controlled, by using a co-integration technology of photonic and electronic circuits in silicon.

The highly flexible and reliable programmability of the qudit processor, that is enabled by technological advances in a mono-lithic integration of all key functionalities and capabilities in a silicon chip, has allowed the implementations of more than one million qudit generators, operators and projectors (see Supplementary Note 8), and also the benchmarking of different generalised quantum algorithms. Such programmability can transition the advanced technologies in controlling qudit states and gates[36–48] to algorithm implementations, playing an enabling role in the roadmap of qudit-based quantum computations. The full chip-scale integration technologies also perfectly match the top-down hierarchy of quantum computing, in which users can define and execute multiple quantum tasks by recompiling the software and reprogramming the quantum hardware. In general, the programmable qudit-based quantum devices can find applications in noise-resilient quantum network[9,10], quantum simulation of complex chemical and physical systems[12–15], and universal quantum computing with qudit cluster states[19–21].

## Data availability

The data that support the plots within this paper and other findings of this study are available from the corresponding author upon reasonable request.

## Code availability

The codes used for the analysis included in the current study are available from the corresponding authors upon reasonable request.

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

## Acknowledgements

We acknowledge X.Wang and S.Tao for useful discussions and assistance of experiment. We acknowledge support from Beijing Natural Science Foundation (Z190005), the National Key R&D Program of China (nos 2019YFA0308702, 2018YFB1107205, 2018YFB2200403, and 2018YFA0704404), the National Natural Science Foundation of China (nos 61975001, 61590933, 61904196, 61675007, and 61775003), and Key R&D Program of Guangdong Province (2018B030329001).

## Author contributions

J.W. conceived the project. Y.C., J.H., Z.C.Z., J.M., Z.N.Z, X.C., C.Z., J.B., T.D., H.Y., M.Z., B.T., and Y.Y. implemented the experiment. Y.C., X.C., and J.B. designed the device. Y.C., J.H., Z.C.Z., J.M. and Z.N.Z provided theoretical analysis. D.D., Z.L., Y.D., L.K.O., M.G.T., J.L.O., Y.L., Q.G., and J.W. managed the project. All authors discussed the results and contributed to the manuscript.

## Competing interests

The authors declare no competing interests.
