## [Peer Review File · Nature Communications]

Reviewers' Comments:

Reviewer #1:

Remarks to the Author:

The authors present a programmable photonic chip that incorporates preparation, manipulation, and measurement. They achieve high fidelities for the logic gates. They implemented the generalised Deutsch-Jozsa and Bernstein-Vazirani algorithms which constitute a proof-of-principle experimental demonstration of a programmable qudit-based processor. They claim that qudit-based quantum computing technology outperform qubit-based quantum technologies and may accelerate the coming of a large-scale quantum computer.

The current manuscript is a much improved version of the first one, I am pleased to see the authors have taken the reviewers' comments on board. The authors now emphasise the programmability of their chip. Programmability within the qudit regime is the main contribution of the work. Monolithic integration of preparation, processing, and detection is also a contribution, and is indeed impressive albeit expected considering the technological advance demonstrated in Ref 35.

The introduction is now representative of the state of the art and the gap that the current work is trying to fill is clearer compared to the first version. As the authors already point out in the reply and the introduction, programmability in qubit technologies is an important enabling step, hence, it is natural to expect the same for qudit technologies. This is what I meant when I said in my previous review that the current work is a "conceptually straightforward next step". I emphasise I agree with the authors that programmability is an enabling step, and it is good to see this is now highlighted. The addition of Figure 1 which has also made this contribution much clearer.

I appreciate the time and effort that the authors invested in adding Supplementary Notes 3 (to explain their scheme), 8 (which counts the resources), and 9 (which analyses the scalability) in response to my suggestions. I am satisfied with the analysis of the scaling, although I note that "techniques of generating multi-photon qudit GHZ states" is probably challenging to do all on-chip. The authors mentioned that their two-ququart processor has six orders of magnitude better detection rate than the corresponding four-qubit device in Ref. 57. The detection rate is dependent on many things and a comparison between Ref. 57 and the current work is not as straightforward especially when you consider that the pump have different powers (a quick read says 4.5 mW average power in Ref. 57 while the current work uses 100 mW power). I suggest a more detailed accounting or a statement to qualify a comparison such as this.

In summary, the work in its current form much improved in its presentation and has more conceptual and technical details that makes the paper an engaging read. (I enjoyed reading it the second time!). I think it is an important work in experimental high-dimensional quantum information and certainly a technological advance that is a step towards a larger scale qudit-based quantum computer.

Some other minor corrections:

Abstract: ...showing great potential... (instead of "potentials")

Introduction, p 2, 1st col: ...qudit states and gates is lacking...(instead of "however lacked")

Results, p 4, 2nd col: ... Fourier gate to obtain the desired solution... (instead of "desire solution")

Results, p 6, 1st col: ...at the expense of repeating m-sequences... (instead of "at the loss of repeating")

Discussion, p 8, 1st col: ...quantum parallelism in terms of... (instead of "in respect of")

Supplementary Note 3, p 4: ... gate to get the desired solution... (instead of "desires solution")

Supplementary Note 3, p 4: ... each photon locally evolves... (instead of "evolutes", I suspect the authors are not referring to the mathematical curve that is an evolute)

Signed: Jacqueline Romero

Reviewer #2:

Remarks to the Author:

I very much appreciated the efforts by the authors to clarify and explain the points raised in the previous revision.

The new version includes a detailed analysis of the scalability and potentiality of such a qudit-based computation scheme. Furthermore, I appreciate that the article now clearly states that this is a proof-of-principle experiment. The technological challenges to scale the size of this processor are still significant. They regard primarily, on the hardware side, the generation of multi-photon GHZ states, and on the software, the control of a large number of optical components in the reconfigurable chip. Although these issues have prevented the realization of a larger qudit-processor at this stage, the authors have individuated a precise roadmap to overcome such limitations in future steps.

Notwithstanding, I found the results a significant technological advancement in qudit processing, and I recommend the publication in Nature Communications.

I have only one last comment regarding some sentences in the discussion and introduction sections. For example, in the Discussion

"Our experimental results show that qudit-based quantum computation can enhance quantum parallelism in respect of the computational capacity, accuracy and efficiency, in comparison with its qubit-based quantum computing counterpart".

It may sound that the various advantages of using qudits instead of qubits have been stated or discovered for the first time in this paper. The enhancement in computational capacity, accuracy, and efficiency along with a more favorable scaling of the Hilbert space dimension in respect of the number of photons is well known in the literature. The authors may consider rephrasing such kinds of sentences to avoid any misleading about the claim of their work.

Reviewer #1:

The authors present a programmable photonic chip that incorporates preparation, manipulation, and measurement. They achieve high fidelities for the logic gates. They implemented the generalised Deutsch-Jozsa and Bernstein-Vazirani algorithms which constitute a proof-of-principle experimental demonstration of a programmable qudit-based processor. They claim that qudit-based quantum computing technology outperform qubit-based quantum technologies and may accelerate the coming of a large-scale quantum computer.

The current manuscript is a much improved version of the first one, I am pleased to see the authors have taken the reviewers' comments on board. The authors now emphasise the programmability of their chip. Programmability within the qudit regime is the main contribution of the work. Monolithic integration of preparation, processing, and detection is also a contribution, and is indeed impressive albeit expected considering the technological advance demonstrated in Ref 35.

The introduction is now representative of the state of the art and the gap that the current work is trying to fill is clearer compared to the first version. As the authors already point out in the reply and the introduction, programmability in qubit technologies is an important enabling step, hence, it is natural to expect the same for qudit technologies. This is what I meant when I said in my previous review that the current work is a "conceptually straightforward next step". I emphasise I agree with the authors that programmability is an enabling step, and it is good to see this is now highlighted. The addition of Figure 1 which has also made this contribution much clearer.

I appreciate the time and effort that the authors invested in adding Supplementary Notes 3 (to explain their scheme), 8 (which counts the resources), and 9 (which analyses the scalability) in response to my suggestions. I am satisfied with the analysis of the scaling, although I note that "techniques of generating multi-photon qudit GHZ states" is probably challenging to do all on-chip. The authors mentioned that their two-ququart processor has six orders of magnitude better detection rate than the corresponding four-qubit device in Ref. 57. The detection rate is dependent on many things and a comparison between Ref. 57 and the current work is not as straightforward especially when you consider that the pump have different powers (a quick read says 4.5 mW average power in Ref. 57 while the current work uses 100 mW power). I suggest a more detailed accounting or a statement to qualify a comparison such as this.

In summary, the work in its current form much improved in its presentation and has more conceptual and technical details that makes the paper an engaging read. (I enjoyed reading it the second time!). I think it is an important work in experimental high-dimensional quantum information and certainly a technological advance that is a step towards a larger scale qudit-based quantum computer.

We are again grateful to the reviewer for her time and expertise in reviewing our manuscript. We are pleasant that the referee has now satisfied our significant modifications of the manuscript, especially the clarifications of technological and conceptual advances, and additional analysis and discussions in Supplementary Notes. Her comments and suggestions have been invaluable in allowing us to clarify and strengthen our manuscript.

We agree with the referee that the comparison of photon detection rate in our work and Ref.57 (now Ref.58) should take the consideration of pump power and many other things. Indeed, the detection rate of photons of any integrated photonic quantum device is dependent on the initial photon-pair generation rate (probability, efficiency) of the parametric SFWM or SDPC sources which relies on the pump power as the referee suggested and also the design of the sources (e.g, microrings, waveguides), loss of on-chip optical component (e.g, fiber coupler, MZ interferometer, beamsplitter, waveguide

crosser, and loss of optical waveguide), and loss of off-chip apparatuses for the measurement and detection of single-photon states (e.g, single-photon detector, optical filter). That is being said, a comprehensive comparison of photon detection rate in our work and in Ref.57 requires much more experimental details. We therefore agree with the referee and we have now added a statement in the main text and discussions in the Supplementary Notes, in order to avoid misleading.

In Page 3 of the main text, we have added a statement as:

ququart device, which is six orders higher than that in a four-qubit device (note the detection rate depends on the performance and loss of the quantum devices as well as their pumping and measurement apparatuses)⁵⁸. Details of device fabrication, state evolution and ex-

In Supplementary Note 1, we have added more detailed discussions as:

higher power excitation), which is 6 orders higher than the four-photon rate of \sim mHz in a device with the same size of Hilbert space as the two-ququarts device here⁵⁸. Note that, the comparison of photon detection rate should take the consideration the performance and loss of the quantum devices as well as their pumping and measurement apparatuses. The detection rate is dependent on the photon-pair generation rate (probability, efficiency) of the parametric sources which relies on the pump power and also the source designs (e.g, microrings or waveguides), loss of on-chip optical components (e.g, chip-fiber coupler, MZI, beamsplitter, waveguide crosser, and loss of optical waveguide), and loss of off-chip apparatuses for the measurement and detection of single-photon states (e.g, single-photon detector, optical filter). That is being said, a comprehensive comparison of photon detection rate requires much more experimental details. In our experiment, we

Some other minor corrections:

Abstract: ...showing great potential... (instead of "potentials")

Introduction, p 2, 1st col: ...qudit states and gates is lacking...(instead of "however lacked")

Results, p 4, 2nd col: ... Fourier gate to obtain the desired solution... (instead of "desire solution")

Results, p 6, 1st col: ...at the expense of repeating m-sequences... (instead of "at the loss of repeating")

Discussion, p 8, 1st col: ...quantum parallelism in terms of... (instead of "in respect of")

Supplementary Note 3, p 4: ... gate to get the desired solution... (instead of "desires solution")

Supplementary Note 3, p 4: ... each photon locally evolves... (instead of "evolutes", I suspect the authors are not referring to the mathematical curve that is an evolute)

We thank the reviewer very much for pointing these corrections. We have now corrected all of these points in the final version of manuscript.

Reviewer #2:

I very much appreciated the efforts by the authors to clarify and explain the points raised in the previous revision. The new version includes a detailed analysis of the scalability and potentiality of such a qudit-based computation scheme. Furthermore, I appreciate that the article now clearly states that this is a proof-of-principle experiment. The technological challenges to scale the size of this processor are still significant. They regard primarily, on the hardware side, the generation of multi-photon GHZ states, and on the software, the control of a large number of optical components in the reconfigurable chip. Although these issues have prevented the realization of a larger qudit-processor at this stage, the authors have individuated a precise roadmap to overcome such limitations in future steps. Notwithstanding, I found the results a significant technological advancement in qudit processing, and I recommend the publication in Nature Communications.

I have only one last comment regarding some sentences in the discussion and introduction sections. For example, in the Discussion "Our experimental results show that qudit-based quantum computation can enhance quantum parallelism in respect of the computational capacity, accuracy and efficiency, in comparison with its qubit-based quantum computing counterpart". It may sound that the various advantages of using qudits instead of qubits have been stated or discovered for the first time in this paper. The enhancement in computational capacity, accuracy, and efficiency along with a more favorable scaling of the Hilbert space dimension in respect of the number of photons is well known in the literature. The authors may consider rephrasing such kinds of sentences to avoid any misleading about the claim of their work.

We are very grateful to the reviewer for his/her time and expertise in reviewing our manuscript. The insightful comments and suggestions on the scalability of our qudit quantum computing scheme have been extremely helpful in allowing us to clarify and further strengthen our manuscript. We also agree with the referee that our current work is a proof-of-principle demonstration of qudit-based quantum computing, which scaling requires future technological progress. We also agree that the presentation of "computational capacity, accuracy and efficiency" in the Discussion and Introduction sections could result in possible misleading and misguidances. We have now rephrased those sentences throughout the paper, as the referee suggested.

In Abstract, we have rephrased the sentence as:

in the processor. Our work shows an integrated photonic quantum technology for qudit-based quantum computing with enhanced capacity, accuracy, and efficiency, which could lead to the acceleration of building a large-scale quantum computer.

In Introduction, we have rephrased the sentence as:

Our results show a proof-of-principle demonstration of qudit-based quantum computer with integrated optics, that allows improvement of the capacity, accuracy and efficiency of quantum computing.

In Conclusion, we have rephrased the sentence as:

Discussion

We have reported a proof-of-principle experimental demonstration of a programmable qudit-based quantum processor in photonic integrated circuits, and implementations of several generalised d -ary quantum Fourier transform algorithms in the d -QPU chip. In agreement with the literature^{17–19,24,43–46}, our experimental results show that qudit-based quantum computation with integrated photonics can enhance quantum parallelism in terms of the computational capacity, accuracy and efficiency, in comparison with its qubit-based quantum computing counterpart. The computational capacity of the